# A Case of Phage Therapy against Pandrug-Resistant *Achromobacter xylosoxidans* in a 12-Year-Old Lung-Transplanted Cystic Fibrosis Patient

**DOI:** 10.3390/v13010060

**Published:** 2021-01-05

**Authors:** David Lebeaux, Maia Merabishvili, Eric Caudron, Damien Lannoy, Leen Van Simaey, Hans Duyvejonck, Romain Guillemain, Caroline Thumerelle, Isabelle Podglajen, Fabrice Compain, Najiby Kassis, Jean-Luc Mainardi, Johannes Wittmann, Christine Rohde, Jean-Paul Pirnay, Nicolas Dufour, Stefan Vermeulen, Yannick Gansemans, Filip Van Nieuwerburgh, Mario Vaneechoutte

**Affiliations:** 1Université de Paris, F-75006 Paris, France; isabelle.podglajen@aphp.fr (I.P.); compain.fabrice@wanadoo.fr (F.C.); jean-luc.mainardi@aphp.fr (J.-L.M.); 2Service de Microbiologie, Unité Mobile d’Infectiologie, AP-HP, Hôpital Européen Georges Pompidou, 20 rue Leblanc, 75015 Paris, France; 3Laboratory Molecular and Cellular Technology, Queen Astrid Military Hospital, Bruynstraat 1, B-1120 Brussels, Belgium; maya.merabishvili@gmail.com (M.M.); jean-paul.pirnay@mil.be (J.-P.P.); 4Laboratory Bacteriology Research, Faculty of Medicine & Health Sciences, Ghent University, C. Heymanslaan 10, B-9000 Gent, Belgium; Leen.VanSimaey@UGent.be (L.V.S.); Hans.Duyvejonck@HOGent.be (H.D.); Mario.vaneechoutte@ugent.be (M.V.); 5Service de Pharmacie, Hôpital européen Georges Pompidou, Assistance Publique-Hôpitaux de Paris Centre Université-Paris, 20 rue Leblanc, 75015 Paris, France; eric.caudron@aphp.fr; 6Lipides, Systèmes Analytiques et Biologiques, Université Paris-Saclay, 92296 Châtenay-Malabry, France; 7CHU Lille, Institut de Pharmacie, F-59000 Lille, France; damien.lannoy@chru-lille.fr; 8ULR7365—GRITA—Groupe de Recherche Sur Les Formes Injectables et Les Technologies Associées, Universit Lille, F-59000 Lille, France; 9Research Center Health & Water Technology, University College Ghent, Keramiekstraat 80, B-9000 Gent, Belgium; Stefan.Vermeulen@HOGent.be; 10Service d’Anesthésie-Réanimation, Hôpital Européen Georges Pompidou, 75015 Paris, France; romain.guillemain@aphp.fr; 11Pediatric Pulmonology and Allergy Unit, Hôpital Jeanne de Flandre, University Lille, CHU Lille, F-59000 Lille, France; caroline.thumerelle@chru-lille.fr; 12Service de Microbiologie, AP-HP, Hôpital Européen Georges Pompidou, 20 rue Leblanc, 75015 Paris, France; 13Unité d'Hygiène Hospitalière, Service de Microbiologie, Hôpital Européen Georges Pompidou, AP-HP, 75015 Paris, France; najiby.kassis-chikhani@aphp.fr; 14Leibniz Institute DSMZ—German Collection of Microorganisms and Cell Cultures GmbH, Inhoffenstraße 7B, 38124 Braunschweig, Germany; jow12@dsmz.de (J.W.); Christine.Rohde@dsmz.de (C.R.); 15Service de Réanimation Médico-Chirurgicale, Centre Hospitalier René Dubos, 95300 Pontoise, France; nicolas.dufour@ght-novo.fr; 16Laboratory of Pharmaceutical Biotechnology, Faculty of Pharmaceutical Sciences, Ghent University, Ottergemsesteenweg 460, B-9000 Gent, Belgium; Yannick.Gansemans@UGent.be (Y.G.); Filip.VanNieuwerburgh@ugent.be (F.V.N.)

**Keywords:** cystic fibrosis, lung transplantation, antibiotic resistance, *Achromobacter xylosoxidans*, bacteriophage therapy

## Abstract

Bacteriophages are a promising therapeutic strategy among cystic fibrosis and lung-transplanted patients, considering the high frequency of colonization/infection caused by pandrug-resistant bacteria. However, little clinical data are available regarding the use of phages for infections with *Achromobacter xylosoxidans*. A 12-year-old lung-transplanted cystic fibrosis patient received two rounds of phage therapy because of persistent lung infection with pandrug-resistant *A. xylosoxidans*. Clinical tolerance was perfect, but initial bronchoalveolar lavage (BAL) still grew *A. xylosoxidans*. The patient’s respiratory condition slowly improved and oxygen therapy was stopped. Low-grade airway colonization by *A. xylosoxidans* persisted for months before samples turned negative. No re-colonisation occurred more than two years after phage therapy was performed and imipenem treatment was stopped. Whole genome sequencing indicated that the eight *A. xylosoxidans* isolates, collected during phage therapy, belonged to four delineated strains, whereby one had a stop mutation in a gene for a phage receptor. The dynamics of lung colonisation were documented by means of strain-specific qPCRs on different BALs. We report the first case of phage therapy for *A. xylosoxidans* lung infection in a lung-transplanted patient. The dynamics of airway colonization was more complex than deduced from bacterial culture, involving phage susceptible as well as phage resistant strains.

## 1. Case Presentation

We report the case of a 12-year-old boy (35.5 kg), who received double lung transplantation in 25 March 2017, (at the age of 12) because of cystic fibrosis. Before transplantation, he was colonized with *Aspergillus fumigatus*, *Achromobacter xylosoxidans* (first colonization identified in 2013), and *Pseudomonas aeruginosa*. After transplantation, he experienced acute kidney injury requiring temporary haemodialysis, pulmonary embolism, and persisting airway colonization with *A. fumigatus*. He was discharged on 29 April 2017, with satisfactory respiratory status; he initially required no home oxygen therapy. His immunosuppressive regimen included prednisolone (25 mg/day), tacrolimus (1.5 mg bis in die (b.i.d.) with a target trough concentration of 6–8 ng/mL), mycophenolate mofetil (1 g b.i.d.) and his antimicrobial prophylaxis was cotrimoxazole (400/80 mg, three times a week), posaconazole (150 mg, oral suspension, ter in die (t.i.d.)), valacyclovir (500 mg b.i.d.), and intravenous immunoglobulins (15 g each three weeks).

Between May and June 2017, he experienced progressive shortness of breath, cough and increased sputum production; he subsequently required the addition of 1–1.5 L/min oxygen therapy. Because of a stenosis of the surgical anastomosis of the main left bronchus, a dilatation during rigid bronchoscopy was performed on 19 May and a Montgomery stent was inserted on 2 June. Later on, a stenosis of the surgical anastomosis on the right bronchus intermedius required a dilatation during rigid bronchoscopy followed by the insertion of an Oki stent on 29 June.

Several bronchoalveolar lavages (BAL), performed between May and June 2017, revealed inflammation (between 800 and >1000 leukocytes/mm^3^ with 88–90% of polymorphonuclear cells) and repeatedly grew pandrug-resistant *A. xylosoxidans* (Table 1 and Appendix A) [1]. Lung biopsy performed on 19 June was consistent with lung infection: diffuse acute bronchiolitis with extension to peri-bronchiolar alveoli. Microbiological samples obtained from lung biopsies were negative (bacterial, mycological, and mycobacteriological cultures). The first antibiotic treatment consisted of intravenous tigecycline 50 mg b.i.d. from 29 June to 31 July. As BAL remained positive for *A. xylosoxidans* and the patient still required oxygen therapy; antibiotic treatment was subsequently switched to imipenem (1200 mg, t.i.d., ~100 mg/Kg/d) on 31 July 2017. Subsequent respiratory samples still grew *A. xylosoxidans* and the respiratory status did not improve (requiring oxygen at home, 1 L/min). Furthermore, despite antibiotic treatment and correct drainage of the right superior lobe, lung consolidation and micronodules remained.

Considering the failure of antibiotic treatment, phage therapy was proposed to the patient and his family, after multidisciplinary discussion. A first cocktail (APC 1.1) containing three lytic phages (JWDelta, JWT and 2-1) active against *A. xylosoxidans* isolate is1S (designated such as later), selected from the DSMZ collection (Braunschweig, Germany) [2,3], was prepared at the Laboratory for Bacteriology Research (Ghent University, Belgium) and the Laboratory for Molecular and Cellular Technology (Queen Astrid Military Hospital, Brussels, Belgium) using the above-mentioned bacterial isolate, as previously described [4]. The production was completed on 23 August with a final phage titer of 4 × 10^10^ plaque forming units (pfu)/mL defined by double-agar overlay method [4], an endotoxin level of 4000 EU/mL (ToxinSensor Chromogenic LAL Endotoxin Assay Kit, GenScript, Piscataway Township, NJ, USA) and a pH of 7.3 (in sterile PBS). Regulatory permission for phage importation was obtained on 1 September 2017, from the Agence Nationale de Sécurité du Médicament (ANSM). A first round of phage administration was performed, consisting of 3 nebulizations/day of 5 mL of the filter-sterilized (Sartorius 0.2 µm) phage solution, tenfold diluted in sterile saline on 8 and 9 September, using a vibrating mesh nebulizer (eFLOW rapid, PARI, PARI GmbH, Germany) with a mouth piece. Immediate tolerance was perfect, but no clinical improvement was noted and culture of subsequent respiratory samples remained positive for *A. xylosoxidans* (10^4^ CFU/mL in the BAL of 15 September).

A second cocktail (APC 2.1) was produced, in which phage JWalpha was added to the three lytic phages of cocktail APC 1.1, to improve the therapeutic efficacy. The production was completed on 22 December with a final phage titer of 5 × 10^9^ pfu/mL, an endotoxin level of 1760 EU/mL and a pH of 6.95 (in sterile PBS). Regulatory permission was obtained for phage importation on 16 January 2018 (ANSM). On 23 January, during therapeutic bronchoscopy under general anesthesia, 30 mL of APC2.1, tenfold diluted (in sterile saline) and filter-sterilized, was instilled in each pulmonary lobe through the fibroscope. Immediate tolerance was perfect and the patient was discharged on 24 January with continued phage nebulization at home: three times a day 5 mL of preparation A using a vibrating mesh nebulizer (eFLOW rapid, PARI) with a mouth piece for 14 days until 6 February. Clinical tolerance was perfect again, but the initial clinical status remained unchanged. BAL performed on 8 February 2018, two days after stopping the phage treatment, still grew 10^5^ cfu/mL of *A. xylosoxidans*. Subsequently, the patient’s respiratory condition slowly improved and oxygen therapy was stopped on 15 February 2018. No clinical worsening was observed after imipenem interruption on 16 February 2018. Sputum culture remained positive for *A. xylosoxidans* but with a low bacterial density (10^3^ CFU/mL in March and May 2018, and June 2019) and no *A. xylosoxidans* could be isolated from BALs#10 and #11, sampled on August 2019, and April 2020, respectively (Table 1). The last pulmonary function tests, performed on 17 October 2019, showed best results since lung transplantation, with an FEV1 of 79% and a forced vital capacity of 85%.

## 2. Microbiological Analysis of Eight *A. xylosoxidans* Isolates

As we observed discrepant outcome in our patient (clinical improvement on the one hand but persisting low-grade airway colonization on the other hand), we decided to study the eight available *A. xylosoxidans* isolates (is1 to is9) from the patient’s colonization obtained by random picking from the culture plates from 4 distinct BALs (Table 2). These eight isolates, taken before, during and after the phage therapy period (Table 2), could not be differentiated from each other by means of MALDI-TOF [5] or McRAPD typing [6] and all belonged to the epidemic clone, already described previously among Belgian CF patients [5]. Whole genome sequencing (WGS) and de novo genome assembly for these eight isolates was carried out (Appendix A). The sequencing reads and genome sequences of all isolates have been submitted to the European Nucleotide Archive (ENA) and are available under project number PRJEB39103 (Table 2). (https://www.ebi.ac.uk/ena/browser/view/PRJEB39103).

Total genome length for all eight isolates was 6.44–6.50 Mbp, with a GC content of 67.5%. All eight isolates showed most similarity with reference genome GCF_008432465 (AX1), which corresponds to the type strain of *A. xylosoxidans* subsp. *xylosoxidans* (ATCC 27061^T^, CCUG 12689^T^, DSMZ 10346^T^, LMG 1863^T^, and NCTC 10807^T^), i.e., between 88.38 and 89.60% on the basis of mapped sequence reads, and between 98.71% and 98.76% on the basis of average nucleotide identity (ANI). Indeed, the overall similarity between the genome sequences of the eight isolates was high, i.e., 99.60%, as assessed by digital DNA hybridization, and more than 99.97% according to ANI-analysis. This was confirmed by variant calling where we observed that the isolates differed from each other only for a total of 367 SNPs, whereas a total of 59,452 SNPs were observed between the isolates and the best matching reference genome (AX1) (Figure 1). The susceptibility phenotype of each of the eight isolates was assessed by spot-test against the phage cocktail APC 2.1 and expressed as “S” when susceptible or “R” when resistant (e.g., “is2S” means that isolate number 2 was susceptible to the phage cocktail) (Table 2 and Appendix A). Among these eight isolates, analysis based on variant calling furthermore revealed four closely related but clearly delineated strains, i.e., clones that differ genetically from other clones, with different phage susceptibility phenotype (Table 2 and Figure 1): Strain 1 (Str1), contained isolates 1 and 3 (is1S and is3R), found in BAL#3 (is1S-Str1/BAL#3) and BAL#5 (is3R-Str1/BAL#5), respectively. Strain 2 (Str2) contained isolate 2 (is2S) from BAL#5 (is2S-Str2/BAL#5). Strain 3 (Str3) contained isolates 5, 6, and 9 (is5S, is6S and is9S), which were found in BAL#6 (is5S-Str3/BAL#6, is6S-Str3/BAL#6) and BAL#9 (is9S-Str3/BAL#9). Strain 4 contained isolates 7 and 8 (is7R and is8R), found in BAL#9 (is7R-Str4/BAL#9 and is8R-Str4/BAL#9) (Table 2).

Isolate is1S-Str1/BAL#3 was obtained on 25 July 2017, i.e., 45 days before the first phage therapy session (8 September 2017) and was 100% identical to is3R-Str1/BAL#5 isolated seven days (15 September 2017) after the first phage therapy session (Figure 1). Since both isolates is1S-Str1/BAL#3 and is3R-Str1/BAL#5 are 100% identical, this means that the phage resistance mechanism of isolate is3R-Str1/BAL#5 is not genetically supported (see discussion). Isolates is5S-Str3/BAL#6 and is6S-Str3/BAL#6, both phage susceptible, were obtained one month later (12 October 2017) and belonged to still another strain (Strain 3). Finally, on 8 February 2018, i.e., 16 days after the second phage therapy session (23 January 2018), an additional three isolates were obtained from BAL#9. Of these, isolate is9S-Str3/BAL#9 belonged to the same Strain 3 as isolates is5S-Str3/BAL#6 and is6S-Str3/BAL#6 and was phage susceptible as well, but isolates 7R and 8R belonged to still another strain (Strain 4) and were both resistant to the phage cocktail. Interestingly, the phage resistant Strain 4 isolates (is7R-Str4/BAL#9 and is8R-Str4/BAL#9) contained mutations compared to the other six isolates that could explain the phage resistance observed for these two isolates (Figure 1): when Strain 1 was used as a reference, a C to A transition at position 1803 of the colicin I receptor gene caused a Tyr601->Stop transition. This receptor has been recognized as a phage receptor [7].

We further studied the course of infection by extracting DNA from the three available *A. xylosoxidans*-positive BAL samples (BAL#5, #6 and #9), and by quantifying DNA of the different strains by means of strain-specific qPCRs (Table 3 and Appendix A) that had been developed on the basis of the obtained WGS data. qPCRs for Strain 2 and Strain 4 were only positive for the BALs from which these isolates had been cultured, respectively, BAL#5 and BAL#9. qPCR for Strain 1 was only positive for BAL#9, but not for BAL#5 from which isolate 3R had been cultured. qPCR for Strain 3 was positive for all three BALs, although no isolate belonging to Strain 3 was cultured from BAL#5.

## 3. Discussion and Conclusions

We report the first case of decolonization of pandrug-resistant *Achromobacter xylosoxidans* after a long-term follow-up in a lung-transplanted patient receiving phage therapy. The first actual demonstration and usage of bacteriophages came almost simultaneously from two microbiologists in 1915 (Frederick Twort) and 1917 (Félix d’Hérelle) [8]. After extensive clinical use during the first half of the 20th century, their use has been progressively reduced, mainly due to the discovery, rapid spread, and ease of use of antibiotics for the treatment of infectious diseases [9]. However, bacteriophages gained a new surge of attention, due to the emergence and spread of pandrug-resistant bacteria [8]. Despite decades of clinical experience with bacteriophages in Eastern Europe, very few data are available regarding their clinical efficacy; besides, case-series and few randomized controlled trials (RCT) [10,11,12]. For instance, a RCT assessed the clinical outcome of patients receiving local application of a phage cocktail for the treatment of burn wound infection. This RCT is frequently cited to have shown no clinical benefits, as compared to standard of care. However, the phage titer that was applied (100 pfu/mL) [10], was so extremely low that it precludes to draw any conclusion from this study regarding the usefulness of phage therapy. The limited number of randomized clinical trials has been recently reviewed [13]. Phage therapy has also been used locally, e.g., for the treatment of bone and joint infections [14,15] or intravenously for the treatment of bloodstream infections [16,17]. Considering the high frequency of colonization and infection caused by antibiotic-resistant bacteria among cystic fibrosis and lung-transplanted patients, such as in our patient, the use of bacteriophages appears to be an appealing option in this population [18,19,20]. In the specific case of *A. xylosoxidans*, thus far a single case-report described the favorable outcome after the use of bacteriophages in a cystic fibrosis patient [21].

Our case-report highlights the many difficulties that need to be taken into account before considering the use of bacteriophages in a clinical setting. These include amongst others an important delay from clinical decision to regulatory agency authorization. Most problematic is the difficult interpretation of the clinical outcome after bacteriophage therapy, considering the long-term persisting airway colonization and the possible evolution of different *A. xylosoxidans* strains. The second therapeutic attempt relied on a massive lung instillation (injection of the bacteriophage solution in the patient’s lung during fibroscopy) followed by a two-week course of phage nebulization. We used a vibrating mesh nebulizer for phage delivery as this type of nebulizer has been shown to result in a lower phage titer reduction, as compared to jet nebulizers [22]. BAL#9, performed two weeks after the second round of phage therapy (8 February 2018) still grew 10^5^ CFU/mL of *A. xylosoxidans*, represented by Strains 3 and 4 (according to culture) and by at least Strains 1, 3 and 4 (according to qPCR) (Table 1, Table 2 and Table 3). Sputum cultures remained positive for 18 months, but with only 10^3^ CFU *A. xylosoxidans*/mL, suggesting a possible upper airway colonization, especially since no *A. xylosoxidans* could be isolated from BALs sampled in August 2019, and April 2020. No re-colonisation occurred more than two years after phage therapy was performed and imipenem treatment was stopped. To more precisely understand the discrepant outcome of our patient, another option would have been to search for active phages from the BAL samples throughout the study, but this was not performed.

Despite the observed differences of a total of 367 SNPs between the four strains of eight *A. xylosoxidans* isolates from the patient, their overall similarity suggests that they originate from a single strain that colonized the patient (who was colonized with *A. xylosoxidans* years before the start of the phage therapy). Therefore, it is possible that some variants were present already before lung transplantation and before the start of phage therapy. Although it is not possible to exclude the simultaneous presence of these different strains on the basis of our culture results, because only one or two colonies were picked from each BAL culture, analysis by means of four strain-specific qPCR assays of the three BAL samples that were available for study, indicated that Strains 2 and 4 were probably prevalent at a particular moment of the course of infection, whereas Strains 1 and 3 could be identified during the full course of the infection. Phage resistant Strain 4 may have been selected during phage therapy.

Phage training has been proposed to overcome phage resistance and increase infectivity against a given bacterial population [23]. However, training phages may take weeks, and therefore might be restricted to the treatment of chronic infections. Regarding more acute infections, the establishment of phage collections (such as the DSMZ, which provided bacteriophages for our patient) is of critical importance to provide phages active against a broad spectrum of bacterial isolates.

An interesting point is the observation of two isolates from the same strain (is1S-Str1/BAL#3 and is3R-Str1/BAL#5) that differ regarding their phage susceptibility phenotype despite the absence of genetic support (identical genome). As observed with bacteria where emerging data point out that a resistance phenotype against a given antimicrobial may not be supported by a genotypic substratum [24,25], phage resistance without any change in genotype has also been documented [26]. These genotype/phenotype discrepancies are hypothesized to be possible through epigenetic regulation mechanisms, of which post-transcriptional regulation, which may influence phage receptor expression on the cell surface or phage receptor masking (through capsule synthesis). Further studies are required to better understand the dynamics of in vivo selection of phage-resistant bacteria.

Because the patient’s medical condition improved anyhow, a few weeks after the second round of phage instillation, it is possible that phage therapy selected the phage-resistant *A. xylosoxidans* Strain 4, but that the mutation that caused phage resistance at the same time decreased the virulence of the strain as already described with other bacteria and phages [27,28,29]. Indeed, we observed several mutations which could explain the phage resistance of Strain 4, of which one was especially interesting, since it caused the gain of a stop codon in the colicin I receptor, which has been recognized as a phage receptor in other species [7]. Because in *E. coli*, the colicin I receptor plays a crucial role in iron transport across the outer membrane [7], loss of this function due to the observed mutation may have caused reduced fitness of the *A. xylosoxidans* mutant Strain 4. However, from a clinical point-of-view, one cannot exclude a spontaneous improvement, independently of phage therapy and despite months of fruitless antibiotic treatment, highlighting the dire need of randomized clinical trials in order to more convincingly assess the clinical efficacy of bacteriophages.

In conclusion, we describe the first case of phage therapy for *A. xylosoxidans* lung infection in a lung-transplanted patient. Despite initial persisting airway colonization, the final clinical and microbiological outcome was favorable. Whole genome sequencing of eight *A. xylosoxidans* isolates and strain-specific qPCR experiments performed on BALs sampled before and after phage therapy allowed to more precisely study the dynamics of lung colonization by different isolates, susceptible or resistant to the applied phage cocktails.

## Figures and Tables

**Figure 1 viruses-13-00060-f001:**
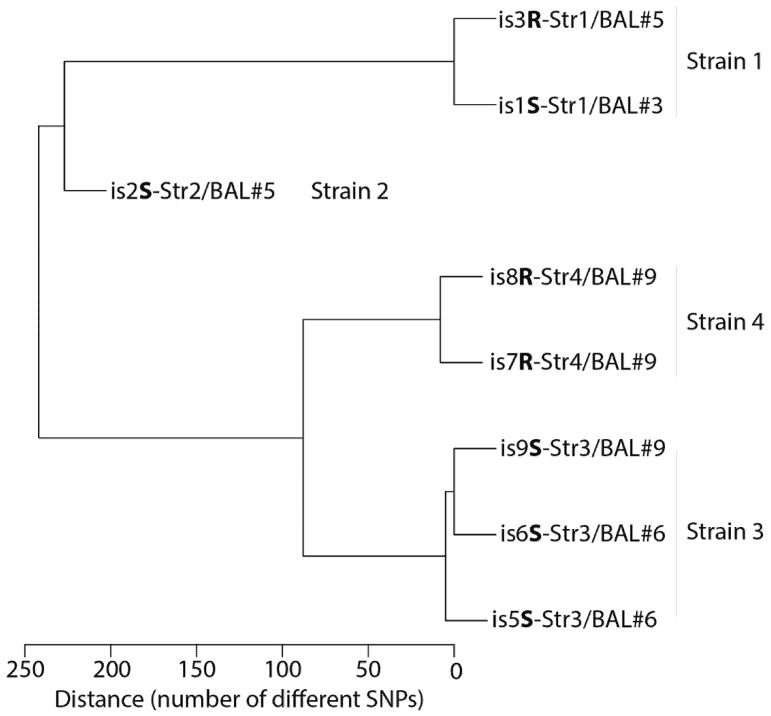
Phylogenetic tree built with Whole genome sequences of eight *Achromobacter xylosoxidans* strains isolated in a 12-year-old boy after double lung transplantation. The distance between samples reflects the number of different Single Nucleotide Polymorphisms (SNPs). Each *A. xylosoxidans* is named as follows: isolate number (is1 to is8) **S** (or **R** to depict susceptibility or resistance to phage cocktail)-Strain number/Number of BAL where it has been identified.

**Table 1 viruses-13-00060-t001:** Results of bronchoalveolar lavages (BALs) and sputum analysis from May 2017, to April 2020, in 12-year-old lung transplanted patient who received two rounds of phage therapy for pandrug-resistant *Achromobacter xylosoxidans* colonization and infection.

Date	19 May 2017	23 May 2017	6 June 2017	12 June 2017	25 July 2017	5 September 2017	8 September 2017	15 September 2017	12 October 2017	27 October 2017	23 January 2018	8 February 2018	5 March 2018	18 May 2018	19 February 2019	19 June 2019	27 August 2019	30 April 2020
**Phage therapy**							**ROUND 1**				**ROUND 2**							
**Tigecycline**				from 29 June to 31 July														
**Imipenem**					From 31 July 2017, to 16 February 2018													
Number of days, relative to the first round of phage therapy	−112	−108	−94	−88	−45	−3		+7	+37	+49	+137	+153	+178	+252	+529	+649	+718	+963
**BAL**																		
BAL Number *	#1			#2	#3	#4		**#5**	**#6**	#7	#8	**#9**			#10		#11	#12
Leukocytes (/mm^3^)	>1000			800	>1000	500		>1000	60	>1000	810	>1000			NA		NA	NA
PMN cells (%)	90			88	90	91		43	30	80	69	82			NA		NA	NA
**Quantification of *A. x* culture (CFU/mL)**	**10^4^**			**10^3^**	**10^4^**	**10^3^**		**10^4^**	**10^3^**	**10^4^**	**10^6^**	**10^5^**			**10^4^**		**0**	**0**
Other bacteria (CFU/mL)	None			None	None	None		None	None	None	None	None			None		10^4^ *S. a*	10^6^ *S. a*
**Sputum**																		
Leucocytes (/field)		>25	>25		>25				>25				NA	NA		NA		
**Quantification of *A. x* culture (CFU/mL)**		**10^5^**	**10^5^**		**10^5^**				**10^6^**				**10^3^**	**10^3^**		**10^3^**		
Other bacteria (CFU/mL)		None	None		None				None				10^8^ *M. c*	None		10^6^ *S. a*		

*A. x: Achromobacter xylosoxidans*; *M. c: Moraxella catarrhalis*; NA: non available; PMN: Polymorphonuclear; RBC: red blood cells; *S. a: Staphylococcus aureus.* *: BAL number in bold indicates BALs for which qPCR had been carried out.

**Table 2 viruses-13-00060-t002:** Eight *Achromobacter xylosoxidans* isolates for which whole genome sequencing was carried out. These were isolated from bronchoalveolar lavages in a 12-year-old boy after double lung transplantation. The patient received a first round of phage administration on 8 September 2017. A second round of phage treatment was performed on 23 January 2018.

Complete Name	Isolate	Strain	Source of Sampling	Date of Sampling	Whole Genome Sequence *
is1S-Str1/BAL#3	is1S	Str1	BAL#3	25 July 2017	ERS5044236-UGAX1
is2S-Str2/BAL#5	is2S	Str2	BAL#5	15 September 2017	ERS5044237-UGAX2
is3R-Str1/BAL#5	is3R	Str1	BAL#5	15 September 2017	ERS5044238-UGAX3
is5S-Str3/BAL#6	is5S	Str3	BAL#6	12 October 2017	ERS5044239-UGAX5
is6S-Str3/BAL#6	is6S	Str3	BAL#6	12 October 2017	ERS5044240-UGAX6
is7R-Str4/BAL#9	is7R	Str4	BAL#9	8 February 2018	ERS5044241-UGAX7
is8R-Str4/BAL#9	is8R	Str4	BAL#9	8 February 2018	ERS5044242-UGAX8
is9S-Str3/BAL#9	is9S	Str3	BAL#9	8 February 2018	ERS5044243-UGAX9

* Submitted at ENA, 1 September 2020. Study ID: PRJEB39103. BAL: Bronchoalveolar lavage, S: Phage susceptible, R: phage resistant.

**Table 3 viruses-13-00060-t003:** Results of bacterial culture (grey-shading) and qPCR of three broncho-alveolar lavages for four different *Achromobacter xylosoxidans* strains isolated in a 12-year-old boy after double lung transplantation.

Bronchoalveolar Lavage		BAL#5	BAL#6	BAL#9
		15 September 2017	12 October 2017	8 February 2018
qPCR for Strain	Isolates			
Str1	is1S, is3R	N	NS	P (34.7)
Str2	is2S	P (36.6)	N	N
Str3	is5S, is6S, is9S	P (32.7)	P (28.9)	P (27.7)
Str4	is7R, is8R	NS	NS	P (35.2)

Grey-shaded box = isolate cultured from this BAL; N: negative (Cq-value > 40); NS: Nonspecific amplification (melting peak different from that of specific amplification); P: positive qPCR (Cq-value).

## Data Availability

The sequencing reads and genome sequences of all isolates have been submitted to the European Nucleotide Archive (ENA) and are available under project number PRJEB39103 (Table 2). (https://www.ebi.ac.uk/ena/browser/view/PRJEB39103).

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
