# Peer review of "A Case of Phage Therapy against Pandrug-Resistant Achromobacter xylosoxidans in a 12-Year-Old Lung-Transplanted Cystic Fibrosis Patient"

_viruses, 2021, doi:10.3390/v13010060_

Round 1

Reviewer 1 Report

Lebeaux et al. report a case study of phage therapy of a cystic fibrosis lung transplant case. Unfortunately, as far as I can tell the only clear indications that phage therapy was successful were clinical improvement and reduction but not elimination of the targeted pathogen from sputum, but not microbiological improvement in the lavages in conjunction with phage treatment until much later. This from my perspective is a near negative result. The successful microbiological reductions months after the termination of phage therapy is too difficult to link to the phage therapy. So the title of the case report is accurate, but that this is an example of successful phage therapy is debatable, as the authors point out near the end of the Discussion. This important concern I think needs to be better stressed throughout the manuscript, that the evidence of success is limited, and better justification needs to be provided for others to consider the results to be indicative that the phage therapy was effective. There also is insufficient discussion of how resistance was being measured and against what phages this resistance was manifest. See the attached marked up PDF for additional comments.

Author Response

Response to Reviewers, “A case of phage therapy against pandrug-resistant Achromobacter xylosoxidans in a 12 year-old lung-transplanted cystic fibrosis patient" by Lebeaux et al (viruses-1038332)

Dear Editor and Reviewers,

We thank you for your helpful and constructive comments on the submitted version of our manuscript and for giving us the opportunity to improve it.

All comments and suggestions have been taken into account as indicated in the provided detailed point-by-point response. All changes made in the revised version of our manuscript are highlighted in yellow.

Reviewer 1:

English language and style

(x) English language and style are fine/minor spell check required

Authors’ response:

All spell corrections were performed (see the highlighted version of the manuscript)

Comments and Suggestions for Authors

Lebeaux et al. report a case study of phage therapy of a cystic fibrosis lung transplant case. Unfortunately, as far as I can tell the only clear indications that phage therapy was successful were clinical improvement and reduction but not elimination of the targeted pathogen from sputum, but not microbiological improvement in the lavages in conjunction with phage treatment until much later. This from my perspective is a near negative result.

Authors’ response:

We agree and modified our manuscript accordingly.

The word ‘success’ has been removed and we present the patient’s outcome in a more balanced way. See for instance lines 108-109: “We report the first case of decolonization of pandrug-resistant Achromobacter xylosoxidans after a long-term follow-up in a lung-transplanted patient receiving phage therapy.”

Furthermore, we also highlighted the pitfalls we met and questions we raise, lines 129-133: “Our case-report highlights the many difficulties that need to be taken into account before considering the use of bacteriophages in a clinical setting. These include amongst others an important delay from clinical decision to regulatory agency authorization. Most problematic is the difficult interpretation of the clinical outcome after bacteriophage therapy, considering the long-term persisting airway colonization and the possible evolution of different A. xylosidans strains.

The successful microbiological reductions months after the termination of phage therapy is too difficult to link to the phage therapy. So the title of the case report is accurate, but that this is an example of successful phage therapy is debatable, as the authors point out near the end of the Discussion.

This important concern I think needs to be better stressed throughout the manuscript, that the evidence of success is limited, and better justification needs to be provided for others to consider the results to be indicative that the phage therapy was effective.

Authors’ response:

As mentioned above, we agree and modified pour manuscript accordingly.

There also is insufficient discussion of how resistance was being measured and against what phages this resistance was manifest.

Authors’ response:

The susceptibility profile of each bacterial isolate was studied by spot-test using the entire phage cocktail. The resistant (R) isolates showed resistance to the overall cocktail, which means to the all of its components together. However, we have not studied the resistance aspects separately for each component of the cocktail Authors’ response:

We added in-depth method description in “Supplementary methods” file and modified our manuscript to describe how was explored the susceptibility of each bacterial isolate to the phage cocktail, lines 63-66 and: The susceptibility phenotype of each of the eight isolates was assessed by spot-test against the phage cocktail APC 2.1 and expressed as “S” when susceptible or “R” when resistant (eg: “is2S” means that isolate number 2 was susceptible to the phage cocktail) (Table 2 and Supplementary Methods).

See the attached marked up PDF for additional comments.

Authors’ response:

We thank the reviewer for careful reading of our manuscript. We addressed all the comments throughout the paper (see highlighted manuscript and “modification table” at the end of this Rebuttal).

Reviewer 2 Report

The aim of the article is to report the case of a lung infection due to pandrug resistant achromobacter xylosoxidans treated with phages and antibiotics in an 12 year-old immunocompromised patient.

Despite microbiological persistence and potential acquisition of phage resistance, the outcome was favorable.

From my point of view, it is crucial to publish such experience, to go ahead and better understand the obstacles and pittfalls that limit the development of phage therapy.

The phages was provided by the DMSZ collection. So it is important to discuss the importance of phage banking.

Authors reported the cocktail phage susceptibility and finally classified the strain as susceptible (S) or resistant (R). What was the definition used to do such categorization? Indeed, if a resistance to one or two phage(s) of the cocktail appeared, was the strain categorize as R? 

Is there any chance to develop phage training to enhance phage activity on achromobacter xylosoxidans that became resistant to the phage cocktail? please also discuss this point, and especially its potential feasability.

Finally in the conclusion, is it correct to mention that microbiological outcome was favorable, whereas A. xylosoxidans persisted in the airway?

Author Response

Response to Reviewers, “A case of phage therapy against pandrug-resistant Achromobacter xylosoxidans in a 12 year-old lung-transplanted cystic fibrosis patient" by Lebeaux et al (viruses-1038332)

Dear Editor and Reviewers,

We thank you for your helpful and constructive comments on the submitted version of our manuscript and for giving us the opportunity to improve it.

All comments and suggestions have been taken into account as indicated in the provided detailed point-by-point response. All changes made in the revised version of our manuscript are highlighted in yellow.

Reviewer 2

Comments and Suggestions for Authors

The aim of the article is to report the case of a lung infection due to pandrug resistant Achromobacter xylosoxidans treated with phages and antibiotics in a 12 year-old immunocompromised patient.

Despite microbiological persistence and potential acquisition of phage resistance, the outcome was favorable.

From my point of view, it is crucial to publish such experience, to go ahead and better understand the obstacles and pittfalls that limit the development of phage therapy.

Authors’ response:

We agree that we encountered several pitfalls during this investigation. These pare listed lines 129-133: “Our case-report highlights the many difficulties that need to be taken into account before considering the use of bacteriophages in a clinical setting. These include amongst others an important delay from clinical decision to regulatory agency authorization. Most problematic is the difficult interpretation of the clinical outcome after bacteriophage therapy, considering the long-term persisting airway colonization and the possible evolution of different A. xylosidans strains.

We also highlighted possible future challenges and research, lines 157-161: “Phage training has been proposed to overcome phage resistance and increase infectivity against a given bacterial population [23]. However, training phages may take weeks, and therefore might be restricted to the treatment of chronic infections. Regarding more acute infections, the establishment of phage collections (such as the DSMZ, which provided bacteriophages for our patient) is of critical importance to provide phages active against a broad spectrum of bacterial isolates.”

And lines 169-170: “Further studies are required to better understand the dynamics of in vivo selection of phage-resistant bacteria.”

The phages was provided by the DMSZ collection. So it is important to discuss the importance of phage banking.

Authors’ response:

we agree and modified the manuscript accordingly, lines 159-161: “Regarding more acute infections, the establishment of phage collections (such as the DSMZ, which provided bacteriophages for our patient) is of critical importance to provide phages active against a broad spectrum of bacterial isolates.”

Authors reported the cocktail phage susceptibility and finally classified the strain as susceptible (S) or resistant (R). What was the definition used to do such categorization? Indeed, if a resistance to one or two phage(s) of the cocktail appeared, was the strain categorize as R?

Authors’ response:

The susceptibility profile of each bacterial isolates was studied by spot-test using the entire phage cocktail. The resistant (R) isolates showed resistance to the overall cocktail, which means to the all of its components together. However, we have not studied the resistance aspects separately for each component of the cocktail as our primary goal was to propose a cure to the patient with a phage cocktail.

We added method description in “Supplementary methods” file and modified our manuscript to describe how was explored the susceptibility of each bacterial isolate to the phage cocktail, lines 63-66 and: The susceptibility phenotype of each of the eight isolates was assessed by spot-test against the phage cocktail APC 2.1 and expressed as “S” when susceptible or “R” when resistant (eg: “is2S” means that isolate number 2 was susceptible to the phage cocktail) (Table 2 and Supplementary Methods).

Is there any chance to develop phage training to enhance phage activity on Achromobacter xylosoxidans that became resistant to the phage cocktail? please also discuss this point, and especially its potential feasability.

Authors’ response:

We agree, and modified the manuscript accordingly, lines 157-159: “Phage training has been proposed to overcome phage resistance and increase infectivity against a given bacterial population [23]. However, training phages may take weeks, and therefore might be restricted to the treatment of chronic infections.”

However, phage training is considered as an efficient (time and feasibility wise) procedure if phage produces at least one single plaque on the bacterial isolate. An active phage mutant can be isolated from that one single plaque, further adapted, and propagated on the given bacterial isolate. The resistant (R) isolates we observed during this study showed complete resistance (no single plaque could be observed) towards the overall cocktail. In this case, it would be more appropriate to choose a new phage, rather than applying adaptation procedure. However, at that stage we decided to discontinue phage treatment.

Finally in the conclusion, is it correct to mention that microbiological outcome was favorable, whereas A. xylosoxidans persisted in the airway?

Authors’ response:

Considering Reviewer 1 and 2’s comments, we modified our manuscript to describe the outcome of bacteriophage therapy in a more balanced way.

However, our conclusion is: “In conclusion, we describe the first case of phage therapy for A. xylosoxidans lung infection in a lung-transplanted patient. Despite initial persisting airway colonization, the final clinical and microbiological outcome was favourable.”

Considering the data we provide, we consider that the final outcome was indeed favorable
